# Novel Usefulness of Krebs von den Lungen 6 (KL-6) with Hemoglobin and Lactate Dehydrogenase for Assessing Bone Marrow Fibrosis

**DOI:** 10.3390/diagnostics12030628

**Published:** 2022-03-03

**Authors:** Minjeong Nam, Mina Hur, Mikyoung Park, Hanah Kim

**Affiliations:** 1Department of Laboratory Medicine, Korea University Anam Hospital, Seoul 02841, Korea; blueccoma@gmail.com; 2Department of Laboratory Medicine, Konkuk University School of Medicine, Seoul 05030, Korea; md.hkim@gmail.com; 3Department of Laboratory Medicine, Eunpyeong St. Mary’s Hospital, College of Medicine, The Catholic University of Korea, Seoul 06591, Korea; mikyoung.pak@gmail.com

**Keywords:** bone marrow fibrosis, KL-6, M_2_BPGi, Hb, LDH

## Abstract

Bone marrow fibrosis (BMF) is manually assessed by reticulin and trichrome stain of bone marrow (BM) biopsy and graded on a semi-quantitative scale. Krebs von den Lungen 6 (KL-6) and Mac-2 binding protein glycosylation isomer (M_2_BPGi) are known to be associated with lung and liver fibrosis, respectively. We explored the usefulness of KL-6 and M_2_BPGi to assess BMF. A total of 250 patients who underwent BM biopsy with hematologic or non-hematologic diseases were included, and 42 patients with lung and liver diseases were excluded. The patients’ data, including age, sex, diagnosis, white blood cell, hemoglobin (Hb), platelet, and lactate dehydrogenase (LDH) were collected. Measured KL-6 and M_2_BPGi levels were compared with reticulin grade (RG) (grade 0–3). KL-6 levels were significantly elevated with an increase in RG, but M_2_BPGi did not show a significant difference. Hb, LDH, or KL-6 were independent predictors for BMF (odds ratio: 1.96, 2.26, 2.91, respectively), but showed poor predictive ability (area under the curve [AUC] 0.62, 0.61, 0.60, respectively). The combination of Hb, LDH, and KL-6 showed a significantly improved predictive ability for BMF (AUC 0.73; integrated discrimination improvement 0.057; category-free net reclassification improvement 0.625). This is the first study to evaluate the usefulness of KL-6 for assessing BMF. The combination of Hb, LDH, and KL-6 would be an objective and relevant biomarker approach and be applied to risk stratification for BMF.

## 1. Introduction

A wide variety of hematologic and non-hematologic diseases are associated with bone marrow fibrosis (BMF). BMF is characterized by increased reticulin and/or collagen fibers deposition in bone marrow (BM) [1]. The etiology and clinical relevance of fiber deposition are not well understood. A pathological increase of BMF contributes to variable degrees of cytopenia, a leukoerythroblastic feature, hepatomegaly/splenomegaly, and an increase in disease-related morbidity [2]. BMF is associated with the prognosis of hematological malignancies such as primary myelofibrosis (PMF), myelodysplastic syndromes, chronic myeloid leukemia, and myeloproliferative neoplasm (MPN) [3,4,5,6]. Thus, identification and grading of BMF are relevant in assessing disease stages and predicting diseases prognosis.

BMF is accessed manually by the stained reticulin fibers on the BM biopsy according to the European consensus system [7]. This conventional grading system is semi-quantitative (grade 0 to 3), and its interpretation may be subjective. The grading system is further restricted by locational heterogeneity of fibrosis, staining inconsistency, and lack of internal control or positive control staining in a given biopsy [8]. However, in the absence of effective countermeasures, the semi-quantitative grading system by reticulin staining is the only method to predict fibrosis in the BM.

BMF is pathologically increased by circulating pro-inflammatory cytokines such as transforming growth factor-beta (TGF-β), interleukin (IL)-8, and IL-2R, as well as direct cellular contact between hematopoietic stem cells (HSCs) and multipotent stromal cells (MSCs). It results in the expansion of osteoblastic lineage cells and osteoblast differentiation in BM [1]. Several studies have suggested that BM cells and progenitor cells could circulate into diverse organs and differentiate into myofibroblasts or fibrocytes in organs [9,10]. Thus, we hypothesized that BMF would be associated with biomarkers for organ fibrosis.

Krebs von den Lungen-6 (KL-6) is widely distributed in the epithelial cells of type II alveoli [11]. The KL-6 levels are increased in interstitial lung disease (ILD), which is a heterogeneous group of diseases characterized by inflammation and fibrosis of the lung parenchyma [12]. Mac-2 binding protein glycosylation isomer (M_2_BPGi) is secreted by hepatic stellate cells (HSCs) and subsequently activates Kupffer cells to facilitate hepatic fibrogenesis [13]. Previous studies demonstrated that KL-6 and M_2_BPGi are associated with fibrosis in the lung and liver and have a role as diagnostic and prognostic markers [12,14,15,16,17]. A preliminary report performed a method validation of M_2_BPGi to assist in the diagnosis of BMF and monitor the therapeutic effect in a small population [18]. However, no study has investigated the potential role of KL-6 and M_2_BPGi in assessing BMF. Therefore, in this study, we aimed to evaluate the usefulness of KL-6 and M_2_BPGi as a new biomarker to assess BMF. We investigated the association of KL-6 and M_2_BPGi with reticulin grade (RG). We aimed to suggest the optimal cut-off for predicting BMF and a new combination of markers to improve the prediction power for BMF.

## 2. Materials and Methods

### 2.1. Study Population

This retrospective study included 250 serum samples obtained from patients who underwent a BM biopsy with hematologic or non-hematologic diseases between August 2020 and April 2021 at the Konkuk University Medical Center (KUMC), Seoul, Korea or the Yeungnam University Medical Center (YUMC), Daegu, Korea. The samples were collected on the same day as the BM biopsy, and residual samples were collected and checked for assessment. To evaluate the impact of BMF on KL-6 and M_2_BPGi levels, we excluded 42 patients with liver and lung diseases to rule out the impact of liver and lung diseases on KL-6 and M_2_BPGi levels. Patients’ demographic, clinical, and laboratory data including age, sex, diagnosis, white blood cell (WBC) (×10^9^/L), hemoglobin (Hb) (g/dL), platelet (×10^9^/L), and lactate dehydrogenase (LDH) (IU/L) were collected using electronic medical records. The laboratory data were collected on the same day that the BM biopsy was performed. The characteristics of patients are summarized in Table 1. The Institutional Review Board of the KUMC and YUMC reviewed this study protocol and exempted the approval of the study with waived informed consent (KUMC, 2020-07-054; YUMC, 2021-01-052).

### 2.2. Reticulin-Based Bone Marrow Fibrosis Score System

Reticulin staining on a BM biopsy was assessed by a hematopathologist according to the European consensus system grade from 0 to 3 [7]: 0, scattered linear reticulin with no intersections corresponding to normal BM; 1, a loose network of reticulin with many intersections; 2, a diffuse and dense increase of reticulin with extensive intersection and occasional focal bundles of collagen and/or focal osteosclerosis; 3, a diffuse and dense increase of reticulin with extensive intersection and coarse bundles of collagen and significant osteosclerosis. RG 2 or 3 is a major criterion for diagnosing overt PMF and showed a similar survival rate or clinical relevance [19]. Moreover, KL-6 levels according to RG 1 to 3 did not show the statistical difference in this study (data not shown). Thus, we merged RG 2 and RG 3 to represent the KL-6 level in high grade.

### 2.3. Measurement of KL-6 and M_2_BPGi Levels

Residual serum samples were collected and were frozen at −70 °C until use. For testing, frozen serum samples were thawed at room temperature and gently mixed for at least 30 min. KL-6 and M_2_BPGi levels were measured using a two-step sandwich chemiluminescence enzyme immunoassay; the assay was performed on a fully automated HISCL-5000 analyzer (Sysmex Corp., Hyogo, Japan) using the HISCL KL-6 Assay Kit and HISCL M_2_BPGi Assay Kit (Sysmex Corp., Kobe, Japan), according to the manufacturer’s instructions.

For KL-6 testing, after KL-6 in the serum sample reacted with biotinylated anti-KL-6 monoclonal antibodies, the complex bound to streptavidin-coated magnetic particles (MPs). Alkaline phosphatase (ALP)-labeled anti-KL-6 monoclonal antibodies specifically bound to KL-6 antigen on the MPs. ALP on the MPs hydrolyzed substrate to an unstable product that emitted light, measured at 477 nm. The intensity of the emitted light reflected the KL-6 level, and the KL-6 level was obtained with a calibration curve. The analytical measurement range (AMR) of KL-6 was 10 to 6000 U/mL.

For M_2_BPGi testing, after reacting wisteria floribunda agglutinin-coated MPs with M_2_BPGi in the serum sample, ALP-labeled anti-M_2_BPGi monoclonal antibodies specifically bind to M_2_BPGi on MP. The light emitted through the ALP reaction was measured at 477 nm, and the M_2_BPGi level was estimated. The AMR of M_2_BPGi was 0.10 to 20.00 cut-off index (C.O.I.).

### 2.4. Statistical Analysis

Data were presented as numbers (percentage) or the median (interquartile range, IQR) after checking for normal distribution and homogenous variation by the Shapiro-Wilk test. The Kruskal-Wallis test with a post-hoc analysis was used to compare KL-6 or M_2_BPGi levels in RG groups (grade 0 to 3). Univariate and multivariate logistic regression analyses were used to identify predictors of BMF; variables included WBC, Hb, PLT, LDH, KL-6, and M_2_BPGi. With the receiver operating characteristic (ROC) curve analysis, area under the curve (AUC) with 95% confidence interval (CI), the optimal cut-off values, sensitivity, and specificity of Hb, LDH, and KL-6 were obtained to predict BMF. The AUCs of each marker and their combination were compared for predictive power. Higher AUC values were considered to demonstrate better discrimination abilities as follow: 0.60 ≤ AUC < 0.70, poor; 0.70 ≤ AUC < 0.80, fair; 0.80 ≤ AUC < 0.90, good; ≥ 0.90, excellent [20]. The cut-off values were calculated by the Youden maximum index value with equal weight to sensitivity and specificity. The Hb, LDH, and KL-6 levels were dichotomized by cut-off values, and the dichotomized results were compared with the proportion of the RG-positive group using the chi-square test. Reclassification analyses using the net reclassification improvement (NRI) and integrated discrimination improvement (IDI) with 95% CI were used to evaluate the value of factors added to Hb or LDH [21]. If NRI and IDI were significantly greater than zero, the new prediction model was considered to have predictive improvement compared with the old model. All statistical analyses were used by IBM SPSS Statistics (version 20, Armonk, NY, USA) and MedCalc Software (version 20.014, MedCalc Software, Ostend, Belgium). *p*-values < 0.05 were considered statistically significant.

## 3. Results

The median levels of KL-6 in RG 1 and RG ≥ 2 were significantly higher than in RG 0 (RG 0, RG 1, and RG ≥ 2: 143, 175, and 212 U/mL, respectively) (RG 0 vs. RG 1, *p* = 0.028; RG 0 vs. RG ≥ 2, *p* = 0.002; RG 1 vs. RG ≥ 2, *p* = 0.157) (Figure 1a). The increase in the KL-6 level according to RG did not show a statistical significance in individual disease groups (data not shown, all *p* > 0.05). The M_2_BPGi level did not show statistical significance (Figure 1b).

In the multivariate analyses, the Hb, LDH, and KL-6 levels were associated with BMF (odds ratio [OR] = 1.96, *p* = 0.049; OR = 2.26, *p* = 0.024; OR = 2.91, *p* = 0.001, respectively) (Table 2). The ROC curve for predicting BMF showed that the optimal cut-off values of Hb, LDH, and KL-6 were ≤10.4 g/dL, >436 IU/L, and >150 U/mL, respectively. Hb, LDH, and KL-6 showed poor predictive ability for BMF (AUC 0.61, 0.60, and 0.62, respectively). Their combination showed significantly increased, fair predictive ability for BMF (AUC = 0.73; *p* < 0.05 in all comparisons); however, it showed lower sensitivity than KL-6 (57.6% vs. 70.5%, *p* = 0.006) (Figure 2).

Reclassification analyses confirmed that the combination of Hb, LDH, and KL-6 significantly increased the predictive ability for BMF compared with Hb or LDH alone (Table 3). The groups with the above cut-off values in Hb, LDH, and KL-6 showed a higher agreement with the positive RG group compared with Hb, LDH, or KL-6 alone (All^+^, 93.3%; Hb^+^, 74.3%; LDH^+^, 76.8%; KL6^+^, 74.4%) (Figure 3).

## 4. Discussion and Conclusions

This study firstly demonstrated the clinical utility of KL-6 to assess BMF. The higher KL-6 levels were significantly associated with an increase in RG, but not M_2_BPGi levels. Our data showed that Hb, LDH, and KL-6 were independent predictors for BMF but had a poor AUC value (0.61, 0.60, 0.62, respectively). Of note, the combination of Hb, LDH, and KL-6 showed higher AUC values (0.73) to predict BMF and high clinical concordance rates with the RG-positive group (93.3%) than Hb, LDH, or KL-6 alone.

Fibrosis is an important process in normal repair responses but is also seen in pathologic responses disrupting the normal tissue architecture, leading to various life-threatening conditions, such as PMF, ILD, and liver fibrosis [22]. Fibrosis is a highly orchestrated process determined in a sophisticated sequence of cytokines and cellular interactions with distinct and shared mechanisms [9,23]. The factors related to disease susceptibility and provocation are often different. However, the factors related to the fibrotic signaling response and progression to fibrotic disease are shared. Although various cellular precursors in different organs are heterogeneous, cells could be activated by shared core pathways, including TGF-β [24]. TGF-β is extensively involved in the development of BMF, lung fibrosis, and liver fibrosis [25,26,27]. In a previous report, megakaryocytes released TGF-β which stimulatesd fibrosis synthesis and increased TGF-β levels which are also elevated in peripheral blood [27]. Therefore, elevated serum TGF-β could affect the lung and liver and could associate with fibrosis biomarkers such as KL-6 and M2BPGi of the lung and liver. Moreover, we investigated the association between KL-6 and CRP and between KL-6 and M_2_BPGi levels. The KL-6 level was associated with the M_2_BPGi level (correlation coefficient ρ = 0.322, *p* < 0.001), not CRP (correlation coefficient ρ = 0.058. *p* = 0.612). Therefore, we might assume that the KL-6 level might reflect general BMF in a chronic phase, not an acute phase.

Many previous studies have demonstrated that KL-6 is valid for diagnosing pulmonary diseases, assessing pulmonary diseases activity, and predicting prognosis in pulmonary diseases [12,14,15]. The role of KL-6 in the pathogenesis of BMF has not been elucidated. Therefore, in the present study, we investigated the potential of KL-6 to serve as a biomarker for assessing BMF. The median levels of KL-6 were 143, 175, and 212 U/mL in RG 0, RG 1, and RG ≥ 2, respectively, and KL-6 levels between RG 0 and RG 1 and between RG 0 and RG ≥ 2 differed significantly (*p* = 0.028 and *p* = 0.002). When the association between the KL-6 level and RG was analyzed in each RG from 0 to 3, RG 1 to 3 did not show any statistical difference (RG 1 vs. RG 2, *p* = 0.101; RG 1 vs. RG 3, *p* = 0.830; RG 2 vs. RG 3, *p* = 0.499). In addition, the increases in high RGs were relatively limited, although the differences between grades were statistically significant. Therefore, through the present results, KL-6 levels could be applied to confirm the presence of BMF, rather than measuring RG 0 to 3 semi-quantitation.

Multivariate analyses confirmed that Hb, LDH, or KL-6 were independent predictors for BMF; but they showed poor AUC values. However, our results are novel regarding the combined use of Hb, LDH, and KL-6 and the evaluation of the clinical concordance rate with the RG-positive group; the combination of Hb, LDH, and KL-6 improved the prediction of BMF (AUC 0.73) and showed a higher concordance rate (93.3%) with the RG-positive group compared with Hb, LDH, or KL-6 alone (AUC 0.61, 0.60, and 0.62; 74.3%, 76.8%, and 74.4%, respectively) (Table 3 and Figure 3). Anemia is one of the cardinal features of BMF, and nearly 40% of patients with BMF showed Hb < 10.0 g/dL [28], and the LDH level is a biomarker of cell turnover and the degree of clonal myeloproliferation [29]. Thus, the high predictive power in the combination marker might indicate that three biomarkers reflected different disease stages and assessed BMF by adding a diagnostic value. Therefore, the combination Hb, LDH, and KL-6 could reflect the clinical aspect better than Hb, LDH, or KL-6 alone.

The biological function of M_2_BPGi has not been fully elucidated, but it has been introduced as a biomarker for liver fibrosis with various causes, such as viral hepatitis, non-alcoholic fatty liver diseases, autoimmune hepatitis, and biliary atresia [17]. Activated HSCs secreting M_2_BPGi are a major source of myofibroblasts after liver injury, but BM-derived fibrocytes could differentiate into collagen type I producing myofibroblast. TGF-β1 and endogenous lipopolysaccharide recruit BM-derived fibrocytes to an injured liver [30]. However, in this study, the association between M_2_BPGi and BMF did not show statistical significance. Therefore, it is assumed that BMF did not affect HSCs producing M_2_BPGi, although fibrosis caused by liver injury could affect BM-derived cell recruitment.

This study has several limitations. First, BMF is a major criterion for PMF. This study included only three PMF cases (1.4%) and a relatively low sample number of high grades. The association between the KL-6 level and RG in the PMF group was not statistically significance. We could not evaluate the expression of KL-6 using immunohistochemistry (IHC) in the biopsy specimen. However, our study investigated the association between KL-6 level and the general BMF in hematologic and non-hematologic diseases. Second, this study included total patients who underwent a BM biopsy with hematologic or non-hematologic diseases. Regardless of diagnosis, disease activity, or treatment, the association between the KL-6 level and RG that was measured on the BM biopsy on the same day was analyzed. This study may indicate that KL-6 levels are associated with fibrosis, not with each individual disease. Third, somatic mutations of Janus kinases 2 (*Jak2*), myeloproliferative leukemia (*MPL*), and calreticulin (*CALR*) genes are known as a driver mutation of the MPN phenotype with BMF [1,2]. Novel treatments for BMF, including Jak2 inhibitors and transplantation, were introduced [31]. In this study, we did not investigate the association between the KL-6 level and *Jak2*, *MPL*, or *CALR* genes and could not elucidate the relevant mechanisms. Further studies are needed to obtain a better understanding of underlying mechanisms.

In conclusion, this study firstly demonstrated the usefulness of KL-6 in assessing BMF. KL-6 is not specific to diagnose BMF. However, the elevation of the KL-6 level gives us information on the overall systemic condition and could serve as a diagnostic biomarker to confirm the presence of BMF. Moreover, the combination of Hb, LDH, and KL-6 showed a better predicting ability and clinical concordance rate with the RG-positive group than Hb, LDH, or KL-6 alone. KL-6, combined with Hb and LDH, could constitute an objective and reliable biomarker panel assessing BMF to replace conventional subjective RG. Further studies with a large study population would be needed to elucidate our findings.

## Figures and Tables

**Figure 1 diagnostics-12-00628-f001:**
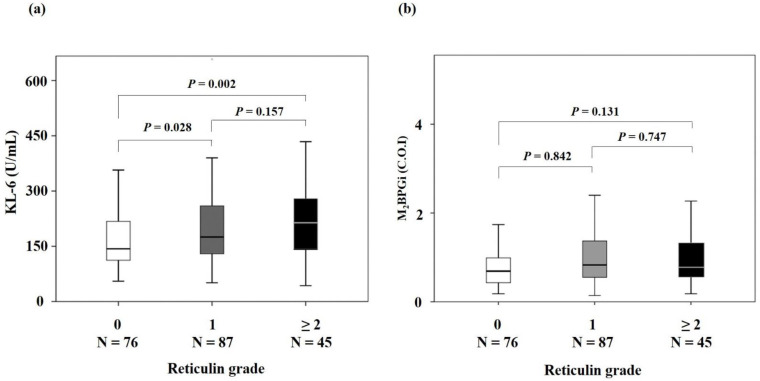
Box and whisker plots for KL-6 (**a**) and M_2_BPGi levels (**b**) at each reticulin grade. In each box plot, a box is drawn from the first quartile to the third quartile, a horizontal line goes through the box at the median, and the whiskers go from each quartile to the minimum or maximum. The KL-6 levels increased with the reticulin grade (*p* = 0.010). There was a significant difference in KL-6 levels between reticulin grade 0 and 1 and between reticulin grade 0 and 2 (*p* = 0.028 and 0.002, respectively; Kruskal-Wallis test with post-hoc analysis). The M_2_BPGi levels, however, did not show significant difference (*p* = 0.060). Abbreviations: KL-6, krebs von den lungen-6; M_2_BPGi, mac-2 binding protein glycosylated isomer; C.O.I, cut-off index.

**Figure 2 diagnostics-12-00628-f002:**
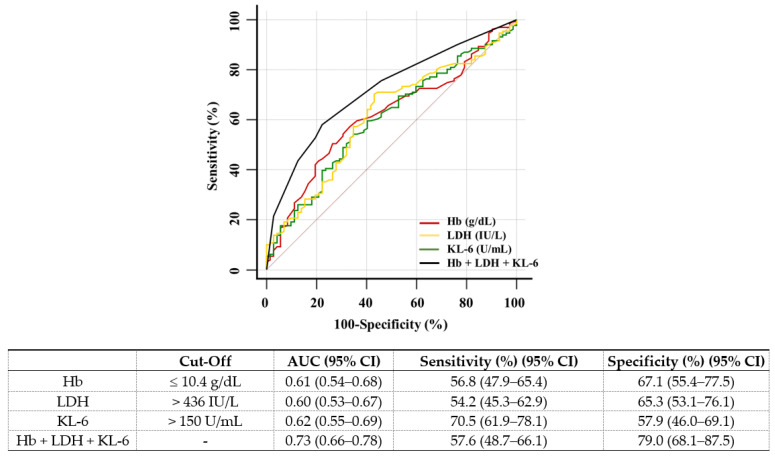
Receiver operator characteristic curve analyses for predicting more than 1 grade of reticulin. Hb, LDH, and KL-6 were comparable to predict more than 1 grade of reticulin (AUC ranging from 0.60 to 0.62); their combination was superior to each marker, showing improved prediction (*p* = 0.006 vs. Hb; *p* = 0.005 vs. LDH; *p* = 0.003 vs. KL-6). Abbreviations: Hb, hemoglobin; LDH, lactate dehydrogenase; KL-6, krebs von den lungen-6.

**Figure 3 diagnostics-12-00628-f003:**
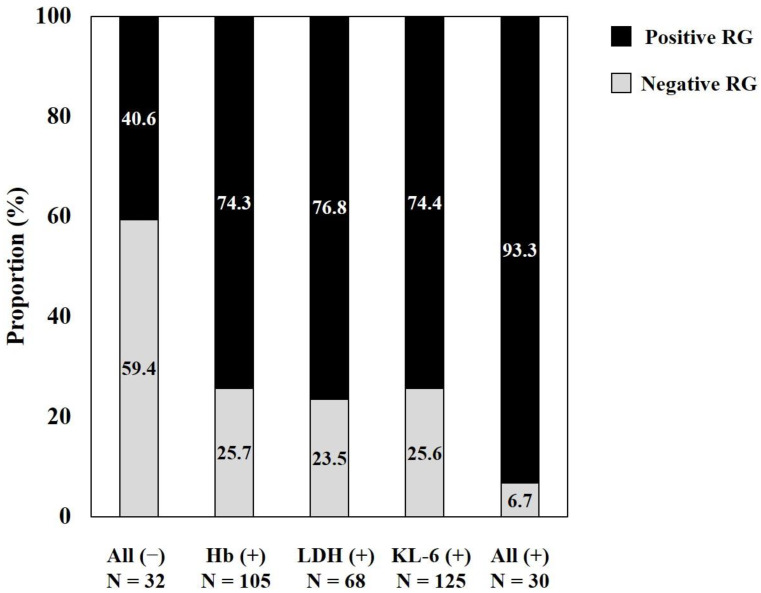
Proportion of RG-positive group according to the dichotomized Hb, LDH, or KL-6 group. All (−) or all (+) indicates a group in which all biomarkers of Hb, LDH, and KL-6 have a value less than or greater than each cut-off value. Hb (+), LDH (+), or KL-6 (+) indicates a group showing a value greater than each cut-off of Hb, LDH, or KL-6 alone. The proportion of RG positive group was 74.3%, 76.8%, and 74.4% in Hb (+), LDH (+), and KL-6 (+) patients, respectively. The proportion of RG positive group increased significantly up to 93.3% when Hb, LDH, and KL-6 were all positive (*p* = 0.026 vs. Hb; *p* = 0.052 vs. LDH; *p* = 0.025 vs. KL-6). Abbreviations: Hb, hemoglobin; LDH, lactate dehydrogenase; KL-6, krebs von den lungen-6; RG, reticulin grade.

**Table 1 diagnostics-12-00628-t001:** Characteristics of the study population.

Characteristics	*n* = 208
Demographic information	
Male	105 (50.5)
Age (years)	59.0 (43.5–69.3)
Clinical information	
Diagnosis	
Lymphoma	47 (22.6)
AML	42 (20.2)
MDS	21 (10.1)
ALL	19 (9.1)
PCM	19 (9.1)
Cytopenia	14 (6.7)
ITP	12 (5.8)
CML	10 (4.8)
ET	5 (2.4)
PMF	3 (1.4)
Others ^a^	16 (7.7)
Laboratory information	
WBC (×10^9^/L)	5.3 (3.0–7.5)
Hb (g/dL)	10.5 (9.4–12.1)
PLT (×10^9^/L)	134.5 (57–229.3)
LDH (IU/L)	431 (351.5–593.0)
BM cellularity (%)	40 (30–70)
Reticulin grade	
Grade 0	76 (36.5)
Grade 1	87 (41.8)
Grade 2	32 (15.4)
Grade 3	13 (6.3)

Data are presented as number a (percentage) or the median (interquartile range). ^a^ Others included lymphocytosis (3), hemophagocytic lymphohistiocytosis (3), erythrocytosis (2), monoclonal gammopathy of undetermined significance (2), mixed phenotype acute leukemia (2), chronic lymphocytic leukemia (1), polycythemia vera (1), prostate cancer (1), and normal (1). Abbreviations: N, number; AML, acute myeloid leukemia; MDS, myelodysplastic syndrome; ALL, acute lymphoblastic leukemia; PCM, plasma cell myeloma; ITP, immune thrombocytopenia; CML, chronic myelogenous leukemia; ET, essential thrombocytosis; PMF, primary myelofibrosis; WBC, white blood cell; Hb, hemoglobin; PLT, platelet; LDH, lactate dehydrogenase; BM, bone marrow.

**Table 2 diagnostics-12-00628-t002:** Logistic regression analyses to identify predictors of bone marrow fibrosis.

	Univariate				Multivariate			
	β	S.E	*p*	OR (95% CI)	β	S.E	*p*	OR (95% CI)
WBC (×10^9^/L)	0.57	0.30	0.056	1.76 (0.99–3.15)	0.40	0.33	0.256	1.49 (0.75–2.98)
Hb (g/dL)	0.96	0.30	0.001	2.62 (1.46–4.70)	0.67	0.32	0.049	1.96 (1.00–3.84)
PLT (×10^9^/L)	0.99	0.30	0.001	2.68 (1.50–4.80)	0.45	0.31	0.204	1.56 (0.79–3.11)
LDH (IU/L)	0.89	0.33	0.007	2.44 (1.27–4.68)	0.82	0.33	0.024	2.26 (1.11–4.61)
KL-6 (U/mL)	1.19	0.30	<0.001	3.28 (1.82–5.91)	1.07	0.33	0.001	2.91 (1.54–5.51)
M_2_BPGi (C.O.I)	0.65	0.33	0.048	1.92 (1.01–3.66)	0.14	0.37	0.702	1.15 (0.56–2.38)

Abbreviations: S.E, standard error; OR, odds ratio; CI, confidence interval; WBC, white blood cell; Hb, hemoglobin; PLT, platelet; LDH, lactate dehydrogenase; KL-6, krebs von den lungen-6; M_2_BPGi, mac-2 binding protein glycosylated isomer; C.O.I, cut-off index.

**Table 3 diagnostics-12-00628-t003:** Evaluation of the added predictive ability of KL-6 using integrated discrimination improvement and net reclassification improvement.

Conventional Marker	Combined Marker	AUC (95% CI)	*p*	IDI		cfNRI	
				Estimated Value (95% CI)	*p*	Estimated Value (95% CI)	*p*
Hb (g/dL)	Hb + KL-6	0.70 (0.64–0.76)	<0.001	0.034 (0.000–0.069)	0.056	0.331 (0004–0.658)	0.048
	Hb + LDH	0.68 (0.61–0.75)	<0.001	0.026 (0.000–0.051)	0.043	0.294 (−0.033–0.622)	0.078
	Hb + LDH + KL-6	0.73 (0.66–0.78)	<0.001	0.044 (0.007–0.081)	0.020	0.434 (0.112–0.756)	0.008
LDH (IU/L)	LDH + KL-6	0.67 (0.60–0.73)	<0.001	0.023 (-0.007–0.054)	0.131	0.364 (0.037–0.692)	0.029
	LDH + Hb	0.68 (0.61–0.75)	<0.001	0.039 (0.022–0.056)	<0.001	0.625 (0.344–0.906)	<0.001
	LDH + Hb + KL-6	0.73 (0.66–0.78)	<0.001	0.057 (0.025–0.090)	0.001	0.625 (0.319–0.931)	<0.001

Abbreviations: AUC, area under the curve, CI, confidence interval; IDI, integrated discrimination improvement; cfNRI, category-free net reclassification improvement; Hb, hemoglobin; KL-6, krebs von den lungen-6; LDH, lactate dehydrogenase.

## Data Availability

The data presented in this study are available from the corresponding author on reasonable request.

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
