# Peer review of "Novel Usefulness of Krebs von den Lungen 6 (KL-6) with Hemoglobin and Lactate Dehydrogenase for Assessing Bone Marrow Fibrosis"

_diagnostics, 2022, doi:10.3390/diagnostics12030628_

Round 1

Reviewer 1 Report

The authors used bone marrow samples and serum samples from patients with various hematology diseases for the study retrospectively. They demonstrated that KL-6 level was associated with bone marrow fibrosis and could be a biomarker. Though interesting, there are some issues in the article.

  1. There are various hematology diseases enrolled in the study, no only the primary marrow fibrosis patients. The authors should rule out that other hematologic malignancies that were enrolled in the study have no impact on KL-6. Especially, the authors showed there is no association between grading of fibrosis and KL-6 level.
  2. The important biomarkers, such as LDH and Hb, are an important items in leukemia and lymphoma which are also enrolled in the study, The authors should also rule out the impact of these diseases on KL-6 level first.
  3. It is a retrospective study that the authors should make sure that the serum they collected before could be check correctly for the assessment the biomarkers they examined in the study. 
  4. The authors should have more experiences of the mechanism of KL-6 on marrow fibrosis for publication.

Author Response

Reviewer #1

Comments to the Author

The authors used bone marrow samples and serum samples from patients with various hematology diseases for the study retrospectively. They demonstrated that KL-6 level was associated with bone marrow fibrosis and could be a biomarker. Though interesting, there are some issues in the article.

  1. There are various hematology diseases enrolled in the study, no only the primary marrow fibrosis patients. The authors should rule out that other hematologic malignancies that were enrolled in the study have no impact on KL-6. Especially, the authors showed there is no association between grading of fibrosis and KL-6 level.

Thank you for your comment. According to the comment, we additionally analyzed the impact of hematologic malignancies on KL-6 level, and most disease groups did not show statistical relevance with KL-6 levels (Rho range, -0.054 – 0.313). In the individual disease group, KL-6 level did not show a statistical difference according to RG (Kruskal-Wallis test, all P > 0.05). We added the following sentences in Results and Discussion sections.

The increase of KL-6 level according to RG did not show a statistical significance in individual disease groups (data not shown, all P > 0.05). (page 9, line 157)

Second, this study included total patients who underwent BM biopsy with hematologic or non-hematologic diseases. Regardless of diagnosis, disease activity, or treatment, the association between KL-6 level and RG that were measured on the BM biopsy on the same day was analyzed. This study may indicate that KL-6 levels are associated with fibrosis, not with each individual disease. (page 12, line 235)

This table show the data that is not shown in the manuscript.

Diagnosis

N (%)

Correlation coefficient

KL-6 according to RG in individual disease group

Rho

P value

P value

Lymphoma

47 (22.6)

-0.051

0.462

0.525

AML

42 (20.2)

0.037

0.599

0.645

MDS

21 (10.1)

0.023

0.739

0.339

ALL

19 (9.1)

-0.069

0.320

0.371

PCM

19 (9.1)

0.313

<0.001

0.052

Cytopenia

14 (6.7)

-0.152

0.029

0.089

ITP

12 (5.8)

-0.091

0.190

0.309

CML

10 (4.8)

-0.003

0.968

0.859

ET

5 (2.4)

-0.029

0.683

0.284

PMF

3 (1.4)

0.086

0.215

0.368

Othersa

16 (7.7)

-0.054

0.437

0.413

  1. The important biomarkers, such as LDH and Hb, are an important item in leukemia and lymphoma which are also enrolled in the study. The authors should also rule out the impact of these diseases on KL-6 level first.

Thank you for your comment. According to your comment, we investigated the impact of lymphoma and leukemia on KL-6 level. As we answered in your comment number 1, lymphoma and leukemia groups were not associated with KL-6 level. KL-6 did not show a statistical difference according to RG in lymphoma and leukemia groups.

  1. It is a retrospective study that the authors should make sure that the serum they collected before could be check correctly for the assessment the biomarkers they examined in the study. 

Thank you for your comment. According to your comment, we modified/added the following sentences in the Method section.

This retrospective study included 250 serum samples obtained from patients who underwent BM biopsy with hematologic or non-hematologic diseases between August 2020 and April 2021 at the Konkuk University Medical Center (KUMC), Seoul or the Yeungnam University Medical Center (YUMC), Daegu, Korea. The samples were collected on the same day as BM biopsy, and residual samples were collected and checked for assessment. (page 6, line 86)

  1. The authors should have more experiences of the mechanism of KL-6 on marrow fibrosis for publication.

Thank you for your comment. According to your comment, we added the following paragraph in the Discussion section.

The factors related to disease susceptibility and provocation are often different. However, the factors related to fibrotic signaling response and progression to fibrotic disease are shared. Although various cellular precursors in different organs are heterogeneous, cells could be activated by shared core pathways, including TGF-β [24]. TGF- β is extensively involved in the development of BMF, lung fibrosis, and liver fibrosis [25-27]. In a previous report, megakaryocytes release TGF-β which stimulates fibrosis synthesis, and increased TGF-β levels are also elevated in peripheral blood [27]. Therefore, elevated serum TGF- β could affect lung and liver and could associate with fibrosis biomarkers of lung and liver such as KL-6 and M2BPGi. Moreover, we investigated the association between KL-6 and CRP and between KL-6 and M2BPGi levels. KL-6 level was associated with M2BPGi level (correlation coefficient ρ = 0.322, P < 0.001), not CRP (correlation coefficient ρ = 0.058. P = 0.612). Therefore, we might assume that KL-6 level might reflect general BMF in a chronic phase, not an acute phase. (page 10, line 184)

In addition, we newly added the following references.

  1. Distler, J.H.W.; Györfi, A.; Ramanujam, M.; Whitfield, M.L.; Königshoff, M.; Lafyatis, R. Shared and distinct mechanisms of fibrosis: Nat. Rev. Rheumatol. 2019, 15, 705-730.

  1. Agarwal, A.;Morrone, K.; Bartenstein, M.; Zhao, Z.J.; Verma, A.; Goel, S. Bone marrow fibrosis in primary myelofibrosis: pathogenic mechanisms and the role of TGF-β. Stem Cell Investig. 2016, 3, 5.

  1. Fernandez, E.E.; Eickelberg, O. The impact of TGF-β on lung fibrosis: from targeting to biomarkers. Proc. Am. Thorac. Soc. 2012, 9, 111-116.

  1. Fabregat, I.; Caballero-Díaz, D. Transforming growth factor-β-induced cell plasticity in liver fibrosis and hepatocarcinogenesis. Front Oncol. 2018, 8, 357.

Reviewer 2 Report

In this paper, Minjeong Nam et al. analyzed the concentration of Krebs von den Lungen 6 (KL-6) and Mac-2 binding protein glycosylation 25 isomer (M2BPGi) in plasma samples derived from patients affected by bone marrow fibrosis. These two proteins have been previously found to be increased in the plasma of patients affected respectively by interstitial lung disease and liver fibrosis and to have a potential role as diagnostic and prognostic markers in these scenarios.

The aim of this study is to understand whether increased levels of these two proteins might assist in the diagnosis and prognosis of bone marrow fibrosis.

The aim of the study is of interest, due to the known difficulties of standardizing the diagnosis and prognosis of bone marrow fibrosis. However, the study presents some limitations, underlined also by authors in the discussion.

  • The patient population analyzed is very heterogeneous, patients affected by different hematologic malignancies have been analyzed. The patients affected by Primary Myelofibrosis are only 3. The analysis in a healthy subject population should be performed.
  • The range of KL-6 plasma values is very wide and, although is statistically significant, the increase in patients with high RG is relatively limited.
  • Is KL-6 produced by bone marrow cells? A mechanistic insight of why a protein produced in the lung is increased in patients affected by bone marrow fibrosis is needed. Authors might perform IHC staining of bone marrow sections.
  • The authors suggest the use of a combination of Hb, LDH and KL-6 to better predict high RG, however, although the AUC is higher in the combination, the sensibility decreased to the level of Hb or LDH alone.
  • Since KL-6 have been previously linked to inflammation, and since bone marrow fibrosis in some circumstances might be a consequence of increased inflammatory cytokines, is the increase of KL-6 also correlated to a marker of inflammation in the cohort of patients analyzed?
  • Are the patients analyzed under therapy? This would represent a variable to be taken into account.

Author Response

Reviewer #2

In this paper, Minjeong Nam et al. analyzed the concentration of Krebs von den Lungen 6 (KL-6) and Mac-2 binding protein glycosylation 25 isomer (M2BPGi) in plasma samples derived from patients affected by bone marrow fibrosis. These two proteins have been previously found to be increased in the plasma of patients affected respectively by interstitial lung disease and liver fibrosis and to have a potential role as diagnostic and prognostic markers in these scenarios.

The aim of this study is to understand whether increased levels of these two proteins might assist in the diagnosis and prognosis of bone marrow fibrosis.

The aim of the study is of interest, due to the known difficulties of standardizing the diagnosis and prognosis of bone marrow fibrosis. However, the study presents some limitations, underlined also by authors in the discussion.

  1. The patient population analyzed is very heterogeneous, patients affected by different hematologic malignancies have been analyzed. The patients affected by Primary Myelofibrosis are only 3. The analysis in a healthy subject population should be performed.

Thank you very much for your valuable comment. Together with the comment No. 1 by the Reviewer 1, we analyzed the correlation between each individual disease and KL-6 level. We agree with your opinion on the limited sample size of PMF. However, we could not increase the sample number due to the patient composition in our hospital. Bone marrow biopsy is a too invasive test to be obtained in the healthy population. Instead, we analyzed the data separately from lymphoma patients without bone marrow involvement (N = 35) and a patient with normal BM (N = 1). In this group, KL-6 level was not associated with BMF (correlation coefficient ρ = 0.111, P = 0.521). The median value of KL-6 did not show a statistical difference according to RG (RG 0, RG 1, RG ≥ 2: 149, 189, 133 U/mL, P = 0.120). According to your comment, we added the following sentences in the Results and Discussion sections.

The increase of KL-6 level according to RG did not show a statistical significance in individual disease groups (data not shown, all P > 0.05). (page 9, line 157)

First, BMF is a major criterion for PMF. This study only included 3 cases (1.4%) of PMF and a relatively low sample number of high grades. The association between KL-6 level and RG in the PMF group did not show statistical significance. However, our study investigated the association between KL-6 and the general BMF in hematologic and non-hematologic diseases. (page 12, line 231)

  1. The range of KL-6 plasma values is very wide and, although is statistically significant, the increase in patients with high RG is relatively limited.

Thank you for your comment. According to your comment, we added the following sentences in Discussion section.

In addition, the increases in high RGs were relatively limited, although the difference between grades were statistically significant. Therefore, through the present results, KL-6 levels could be applied to confirm the presence of BMF, rather than measuring RG 0 to 3 semi-quantitation. (page 11, line 205)

  1. Is KL-6 produced by bone marrow cells? A mechanistic insight of why a protein produced in the lung is increased in patients affected by bone marrow fibrosis is needed. Authors might perform IHC staining of bone marrow sections.

Thank you for your comment. According to your comment, as well as the first reviewer’s comment (comment number 4), we added the following paragraph in the Discussion section. Unfortunately, we could not perform the IHC staining; instead, we added the following sentences in the Discussion section, too.

The factors related to disease susceptibility and provocation are often different. However, the factors related to fibrotic signaling response and progression to fibrotic disease are shared. Although various cellular precursors in different organs are heterogeneous, cells could be activated by shared core pathways, including TGF-β [24]. TGF- β is extensively involved in the development of BMF, lung fibrosis, and liver fibrosis [25-27]. In a previous report, megakaryocytes release TGF-β which stimulates fibrosis synthesis, and increased TGF-β levels are also elevated in peripheral blood [27]. Therefore, elevated serum TGF- β could affect lung and liver and could associate with fibrosis biomarkers of lung and liver such as KL-6 and M2BPGi. Moreover, we investigated the association between KL-6 and CRP and between KL-6 and M2BPGi levels. KL-6 level was associated with M2BPGi level (correlation coefficient ρ = 0.322, P < 0.001), not CRP (correlation coefficient ρ = 0.058. P = 0.612). Therefore, we might assume that KL-6 level might reflect general BMF in a chronic phase, not an acute phase. (page 10, line 184)

We could not evaluate the expression of KL-6 using immunohistochemistry (IHC) in the biopsy specimen. However, our study investigated the association between KL-6 level and the general BMF in hematologic and non-hematologic diseases. (page 12, line 233)

In addition, we newly added the following references.

  1. Distler, J.H.W.; Györfi, A.; Ramanujam, M.; Whitfield, M.L.; Königshoff, M.; Lafyatis, R. Shared and distinct mechanisms of fibrosis: Nat. Rev. Rheumatol. 2019, 15, 705-730.

  1. Agarwal, A.;Morrone, K.; Bartenstein, M.; Zhao, Z.J.; Verma, A.; Goel, S. Bone marrow fibrosis in primary myelofibrosis: pathogenic mechanisms and the role of TGF-β. Stem Cell Investig. 2016, 3, 5.

  1. Fernandez, E.E.; Eickelberg, O. The impact of TGF-β on lung fibrosis: from targeting to biomarkers. Proc. Am. Thorac. Soc. 2012, 9, 111-116.

  1. Fabregat, I.; Caballero-Díaz, D. Transforming growth factor-β-induced cell plasticity in liver fibrosis and hepatocarcinogenesis. Front Oncol. 2018, 8, 357.

  1. The authors suggest the use of a combination of Hb, LDH and KL-6 to better predict high RG, however, although the AUC is higher in the combination, the sensibility decreased to the level of Hb or LDH alone.

Thank you for your comment. According to your comment, we modified the following sentence in the Results section.

Their combination showed significantly increased fair predictive ability for BMF (AUC = 0.73; P < 0.05 in all comparisons); however, it showed lower sensitivity than KL-6 (57.6% vs. 70.5%, P = 0.006) (Figure 2). (page 9, line 164)

  1. Since KL-6 have been previously linked to inflammation, and since bone marrow fibrosis in some circumstances might be a consequence of increased inflammatory cytokines, is the increase of KL-6 also correlated to a marker of inflammation in the cohort of patients analyzed?

Thank you for your comment. According to your comment, we further analyzed the association between KL-6 level and inflammatory markers. We added the following sentences in the Discussion section.

Moreover, we investigated the association between KL-6 and CRP and between KL-6 and M2BPGi levels. KL-6 level was associated with M2BPGi level (correlation coefficient ρ = 0.322, P < 0.001), not CRP (correlation coefficient ρ = 0.058. P = 0.612). Therefore, we might assume that KL-6 level might reflect general BMF in a chronic phase, not an acute phase. (page 10, line 192)

  1. Are the patients analyzed under therapy? This would represent a variable to be taken into account.

Thank you for your comment. We added the following sentences in the Discussion section as a study limitation.

Second, this study included total patients who underwent BM biopsy with hematologic or non-hematologic diseases. Regardless of diagnosis, disease activity, or treatment, the association between KL-6 level and RG that were measured on the BM biopsy on the same day was analyzed. This study may indicate that KL-6 levels are associated with fibrosis, not with each individual disease. (page 12, line 235)

Reviewer 3 Report

In this manuscript, Nam et al. explored the use of KL-6 and M2BPGi serum biomarkers in the context of bone marrow fibrosis assessment. They evaluated 208 patients with available bone marrow biopsy and observed that KL-6 only was significantly related to the Reticulin-based fibrosis assessment. Additionally, the improved predictive ability of KL-6 was observed if combined with hemoglobin and lactate dehydrogenase levels.

Overall, the report requires revision to provide a more accurate description.

Please, address the following comments.

Major points:

1) The etiology and clinical relevance of fibrosis in general and bone marrow fibrosis, in particular, are not unknown. Please, revise the manuscript and modify sentences accordingly (i.e., page 2, lines 48-49).

2) Bone marrow fibrosis is particularly relevant in the context of primary myelofibrosis. However, this condition is poorly represented in the current cohort of cases (1.4% of cases). This limitation should be additionally reported in the manuscript.

3) What are the reasons to associate such different pathologies/diseases in the “Other” category? This is unclear as some of the conditions within the “Other” category are more represented than others specifically reported (e.g., cytopenia with 14 cases, immune thrombocytopenia with 12 cases, …)? Within the “Other” category, what is the specific diagnosis of the “tumor” (page 11, line 380) reported?

4) The reasons behind the excluding criteria (lung and liver diseases) should be clearly stated in the Methods.   

5) It is unclear which Reticulin-based bone marrow fibrosis score system was performed in the study. Please, state it clearly in the Methods section.

6) The World Health Organization suggests performing the 4-tier Reticulin score of the European consensus system (as reported in the Introduction). This score ranges from 0 to 3. If this is the scoring system performed in this study, it should be stated why grade 2 and grade 3 were unified and considered together and not analyzed separately (as for the other grades). Is there a particular (clinical) reason justifying this choice? The correlation analysis of KL-6 and M2BPGi taking into account every single grade of the score adopted and the relative p-values should be presented.

6) p-values should be reported in the Results when describing biomarkers association (page 4 lines 165-166 and page 5 lines 167-168). In particular, KL-6 did not significantly discriminate between scores 1 and ≥ 2 (Figure 1a. p=0.157). This data should be reported in the Result section and thoroughly commented in the Discussion.

7) As M2BPGi did not demonstrate clinical relevance, the first sentence of the Discussion should be revised (page 5, lines 185-186).

8) In the Discussion, data regarding JAK2, MPL, and CALR mutations are abruptly presented (page 6, lines 245-248). However, in the Methods, no molecular analysis is described. Similarly, no data about mutations are reported in the Result section, nor is the correlation between mutations and reticulin grades or biomarkers values. If the molecular analysis was performed, it should be clearly assessed, and its procedure and results thoroughly described and commented.  

Minor points:

9) Revise the Title, providing insights about the usefulness of combining KL-6 with hemoglobin and lactate dehydrogenase.

10) The Manuscript should be revised for English wording and phrasing.

11) Table 1, page 11, lines 350-361: the sum of all the values of the single diagnosis resulted in 209 cases and not 208. Same for the Reticulin grade (76+87+37+13 = 213 cases instead of 208). Please, revise and correct Table 1, providing correct values for each variable.

12) Table 1, page 11: it is unclear the timing of the hematopoietic stem cell transplantation related to the biopsy and in which conditions it was performed. Please, provide more specific information about this variable, if available.

13) Table 3, page 13: please, provide a better heading for the “New marker” column, as the variables reported are not new but combined markers.

Author Response

Reviewer #3

In this manuscript, Nam et al. explored the use of KL-6 and M2BPGi serum biomarkers in the context of bone marrow fibrosis assessment. They evaluated 208 patients with available bone marrow biopsy and observed that KL-6 only was significantly related to the Reticulin-based fibrosis assessment. Additionally, the improved predictive ability of KL-6 was observed if combined with hemoglobin and lactate dehydrogenase levels.

Overall, the report requires revision to provide a more accurate description. Please, address the following comments.

Major points:

  1. The etiology and clinical relevance of fibrosis in general and bone marrow fibrosis, in particular, are not unknown. Please, revise the manuscript and modify sentences accordingly (e., page 2, lines 48-49).

Thank you for the comment. We modified the following sentence in the Introduction section.

The etiology and clinical relevance of fiber deposition are not well understood. (page 4, line 50)

  1. Bone marrow fibrosis is particularly relevant in the context of primary myelofibrosis. However, this condition is poorly represented in the current cohort of cases (1.4% of cases). This limitation should be additionally reported in the manuscript.

Thank you for your comment. According to your comment, we added the following sentences in the Discussion section as a study limitation.

First, BMF is a major criterion for PMF. This study only included three PMF cases (1.4%) and a relatively low sample number of high grades. The association between KL-6 level and RG in the PMF group did not show a statistical significance. We could not evaluate the expression of KL-6 using immunohistochemistry (IHC) in the biopsy specimen. However, our study investigated the association between KL-6 level and the general BMF in hematologic and non-hematologic diseases. (page 12, line 230)

  1. What are the reasons to associate such different pathologies/diseases in the “Other” category? This is unclear as some of the conditions within the “Other” category are more represented than others specifically reported (g., cytopenia with 14 cases, immune thrombocytopenia with 12 cases, …)? Within the “Other” category, what is the specific diagnosis of the “tumor” (page 11, line 380) reported?

Thank you for your comment. According to your comment, cytopenia and ITP belonging to the “other” category were analyzed separately, and we modified Table 1 (page 19, Table 1). We also modified the tumor to the specific diagnosis.

  1. The reasons behind the excluding criteria (lung and liver diseases) should be clearly stated in the Methods. 

Thank you for your comment. According to your comment, we added the following sentence in the Method section.

To evaluate the impact of BMF on KL-6 and M2BPGi levels, we excluded 42 patients with liver and lung diseases to rule out the impact of liver and lung diseases on KL-6 and M2BPGi levels. (page 6, line 90)

  1. It is unclear which Reticulin-based bone marrow fibrosis score system was performed in the study. Please, state it clearly in the Methods section.

Thank you for your comment. According to your comment, we added the following in the Methods section.

Reticulin-based bone marrow fibrosis score system

Reticulin staining on a BM biopsy was assessed by a hematopathologist according to the European consensus system grade from 0 to 3 [7]: 0, scattered linear reticulin with no intersections corresponding to normal BM; 1, a loose network of reticulin with many intersections; 2, a diffuse and dense increase of reticulin with extensive intersection and occasional focal bundles of collagen and/or focal osteosclerosis; 3, a diffuse and dense increase of reticulin with extensive intersection and coarse bundles of collagen and significant osteosclerosis. RG 2 or 3 is a major criterion for diagnosing overt PMF and show similar survival rate or clinical relevance [19]. Moreover, KL-6 levels according to RG 1 to 3 did not show the statistical difference in this study (data not shown). Thus, we merged RG 2 and RG 3 to represent the KL-6 level in high grade. (page 6, line 101)

We also added a new reference.

  1. Mudireddy, M.; Shah, S.; Lasho, T.; Barraco, D.; Hanson, C.A.; Ketterling, R.P.; Gangat, N.; Pardanani, A.; Tefferi, A. Prefibrotic versus overtly fibrotic primary myelofibrosis: clinical, cytogenetic, molecular and prognostic comparisons. Br. J. Haematol. 2018, 182, 594-597.

  1. The World Health Organization suggests performing the 4-tier Reticulin score of the European consensus system (as reported in the Introduction). This score ranges from 0 to 3. If this is the scoring system performed in this study, it should be stated why grade 2 and grade 3 were unified and considered together and not analyzed separately (as for the other grades). Is there a particular (clinical) reason justifying this choice? The correlation analysis of KL-6 and M2BPGi taking into account every single grade of the score adopted and the relative p-values should be presented.

Thank you for your comment. According to your comment, we added the following sentences in the Methods section and Discussion section.

RG 2 or 3 is a major criterion for diagnosing overt PMF and show similar survival rate or clinical relevance [19]. Moreover, KL-6 levels according to RG 1 to 3 did not show the statistical difference in this study (data not shown). Thus, we merged RG 2 and RG 3 to represent the KL-6 level in high grade.  (page 7, line 108)

When the association between KL-6 level and RG were analyzed in every single RG from 0 to 3, RG 1 to 3 did not show statistical difference (RG 1 vs. RG 2, P = 0.101; RG 1 vs. RG 3, P = 0.830; RG 2 vs. RG 3, P = 0.499). (page 11, line 203)

  1. p-values should be reported in the Results when describing biomarkers association (page 4 lines 165-166 and page 5 lines 167-168). In particular, KL-6 did not significantly discriminate between scores 1 and ≥ 2 (Figure 1a. p=0.157). This data should be reported in the Result section and thoroughly commented in the Discussion.

Thank you for your comment. According to your comment, we modified/added the following sentences in the Results and Discussion sections.

The median levels of KL-6 in RG 1 and RG ≥ 2 were significantly higher than in RG 0 (RG 0, RG 1, and RG ≥ 2: 143, 175, and 212 U/mL, respectively) (RG 0 vs. RG 1, P = 0.028; RG 0 vs. RG ≥ 2, P = 0.002; RG 1 vs. RG ≥ 2, P = 0.157) (Figure 1a). (page 9, line 155)

The median levels of KL-6 were 143, 175, and 212 U/mL in RG 0, RG 1, and RG ≥2, respectively, and KL-6 levels between RG 0 and RG 1 and between RG 0 and RG ≥ 2 differed significantly (P = 0.028 and P = 0.002). (page 11, line 201)

  1. As M2BPGi did not demonstrate clinical relevance, the first sentence of the Discussion should be revised (page 5, lines 185-186).

Thank you for your comment. According to your comment, we modified the following sentence in the Discussion section.

This study firstly demonstrated the clinical utility of KL-6 to assess BMF. (page 10, line 174)

  1. In the Discussion, data regarding JAK2, MPL, and CALRmutations are abruptly presented (page 6, lines 245-248). However, in the Methods, no molecular analysis is described. Similarly, no data about mutations are reported in the Result section, nor is the correlation between mutations and reticulin grades or biomarkers values. If the molecular analysis was performed, it should be clearly assessed, and its procedure and results thoroughly described and commented.  

Thank you for your valuable comment. In the study, we collected results of molecular analysis but did not analyze the data due to small numbers of samples. Therefore, we modified the following sentences in the Discussion section as a study limitation.

In this study, we did not investigate the association between KL-6 and Jak2, MPL, or CALR gene and could not elucidate the relevant mechanisms. Further study will be needed to obtain a better understanding of mechanisms. (page 12, line 242)

Minor points:

  1. Revise the Title, providing insights about the usefulness of combining KL-6 with hemoglobin and lactate dehydrogenase.

Thank you for the comment. According to your comment, we modified the title.

Novel Usefulness of Krebs von den Lunge 6 (KL-6) with Hemoglobin and Lactate Dehydrogenase for Assessing Bone Marrow Fibrosis

  1. The Manuscript should be revised for English wording and phrasing.

Thank you. We checked and corrected the English wording and phrasing.

  1. Table 1, page 11, lines 350-361: the sum of all the values of the single diagnosis resulted in 209 cases and not 208. Same for the Reticulin grade (76+87+37+13 = 213 cases instead of 208). Please, revise and correct Table 1, providing correct values for each variable.

Thank you for your comment. According to your comment, we corrected the typo errors in Table 1.

  1. Table 1, page 11: it is unclear the timing of the hematopoietic stem cell transplantation related to the biopsy and in which conditions it was performed. Please, provide more specific information about this variable, if available.

Thank you for the comment. The patient group who underwent HSCT had no effect on the conclusion of the study, so the content was deleted in the modified Table 1.

  1. Table 3, page 13: please, provide a better heading for the “New marker” column, as the variables reported are not new but combined markers.

Thank you for your comment. According to your comment, we modified the column heading using “combined markers” in Table 3.

Round 2

Reviewer 1 Report

The authors respond and modified the manuscript to reviewers' comments accordingly. Therefore, I think it is suitable for publication in the Journal.

Author Response

We appreciate you for your time and effort on our work.

Reviewer 2 Report

Although non experimentally but only by editing the manuscript, the authors addressed my previous concerns.  

Author Response

(The authors gave the same response as above.)

Reviewer 3 Report

The Authors accurately addressed all the comments, but the modifications dampened the overall significance of the study. 

Please, double-check the values of each RG category values reported in Figure 1, as they differ from the value reported in Table 1 and their sum is smaller than the total sample size (76+87+35=198 vs. 208) [Please, note that these values were not reported in Figure 1 at the time of the first version; therefore, they could have not been reviewed].

Author Response

Thank you for your valuable comment. We went through all the numerical values and modified the error in the manuscript. Sorry for the typo error. We noted this change with red color in the figure.
